# Worst-case Performance of Popular Approximate Nearest Neighbor Search Implementations: Guarantees and Limitations

**Piotr Indyk**
MIT
indyk@mit.edu

**Haike Xu**
MIT
haikexu@mit.edu

## Abstract

Graph-based approaches to nearest neighbor search are popular and powerful tools for handling large datasets in practice, but they have limited theoretical guarantees. We study the worst-case performance of recent graph-based approximate nearest neighbor search algorithms, such as HNSW, NSG and DiskANN. For DiskANN, we show that its "slow preprocessing" version provably supports approximate nearest neighbor search query with constant approximation ratio and poly-logarithmic query time, on data sets with bounded "intrinsic" dimension. For the other data structure variants studied, including DiskANN with "fast preprocessing", HNSW and NSG, we present a family of instances on which the empirical query time required to achieve a "reasonable" accuracy is linear in instance size. For example, for DiskANN, we show that the query procedure can take at least $0.1n$ steps on instances of size $n$ before it encounters *any* of the 5 nearest neighbors of the query.

## 1  Introduction

The nearest neighbor search (NN) problem is defined as follows: given a set of $n$ points $P$ in a metric space $(X, D)$, build a data structure that, given any query point $q \in X$, returns $p \in P$ closest to $q$. More generally, given a parameter $k$, the data structure should report $k$ points in $P$ that are closest to $q$. Often, though not always, the metric space is a $d$-dimensional vector space with the distance function induced by the Euclidean norm. Since its introduction in the influential book by Minsky and Papert in the 1960s [31], the problem has found a tremendous number of applications in machine learning and computer vision [34]. In most of those applications, the underlying metric is induced by a set of points in a high-dimensional space. Since in this setting worst-case efficient nearest neighbor algorithms are unlikely to exist (see e.g., [2]), various *approximate* formulations of the problem have been studied extensively. One popular formulation allows the algorithm to return any point $p' \in P$ whose distance to the query $q$ is at most $c$ times the distance between $q$ and its true nearest neighbor in $P$; such point $p'$ is called a *c-approximate nearest neighbor*. In practice, the accuracy of an approximate data structure is often evaluated by estimating the recall, i.e., the average fraction of the true $k$ nearest neighbors returned by the data structure.

Over the last few decades, many nearest neighbor data structures have been proposed, both exact and approximate. The most popular classes of data structures are tree algorithms (e.g., kd-trees [4]); locality sensitive hashing [1]; learning-to-hash and product quantization [35, 36] ; and metric data structures. The last class includes algorithms that work for point-sets in *arbitrary* metrics, not just those in $d$-dimensional vectors normed spaces. This category can be further subdivided into algorithms based on metric trees [6, 32], divide and conquer algorithms [24, 5, 10], and methods based on greedy search in proximity graphs [3, 28, 15, 21]. See the surveys [8, 27] for further overview of these classes of algorithms.

37th Conference on Neural Information Processing Systems (NeurIPS 2023).

Several algorithms in the latter class, such as HNSW [28], NSG [15] and DiskANN [21], are widely used in practice.[1] They have been recently shown empirically to provide excellent tradeoffs between accuracy and query speed [27]. Their design is based on approximating theoretical concepts such as Delaunay or Relative Neighborhood Graphs. However, their worst-case performance is not well understood, especially when the dimension is high. For example, the authors of [28] point out that "Further analytical evidence is required to confirm whether the resilience of Delaunay graph approximations generalizes to higher dimensional spaces." This stands in contrast to the older algorithms, e.g., those based on the divide and conquer approach, which come with worst-case performance guarantees. Those algorithms are typically analyzed under the assumption that the input point set $P$ has low *doubling dimension* (a measure of the intrinsic dimensionality of the point-set)[2] and provide near-linear space and logarithmic query time bounds, with the big-Oh constants depending on the doubling dimension. This raises the question whether similar bounds can be obtained for the newer algorithms. On the practical side, investigating the worst-case behavior of data structures is important to understand their benefits and limitations.

**Our results**   In this paper we initiate the study of the worst-case performance of the recent metric data structures based on proximity graphs. As in [21], we mostly focus on three popular algorithms and their implementations: HNSW, NSG and DiskANN. (In addition, we present similar results for other graph-based algorithms in the supplementary material section.) We present both upper and lower bounds on their worst-case search times. Our specific contributions are as follows:

- **Upper bounds**: For one of the data structures studied, namely DiskANN version with "slow preprocessing", we are able to show a provable worst-case upper bound on its performance. Specifically, we show that (a) the greedy search procedure returns an $\left(\frac{\alpha+1}{\alpha-1} + \epsilon\right)$-approximate neighbor in $O\left(\log_\alpha \frac{\Delta}{(\alpha-1)\epsilon}\right)$ steps and (b) each step takes at most $O((4\alpha)^d \log \Delta)$ time. This implies that the overall running time is poly-logarithmic in $\Delta$ when $d$ is constant. Here $\alpha > 1$ denotes a parameter of the DiskANN algorithm (described in Preliminaries, typically set to 2), $d$ denotes the doubling dimension, while $\Delta$ denotes the *aspect* ratio of the input set $P$, i.e., the ratio between the diameter and the distance of the closest pair [3]. We also show that our approximation bound is tight, and that the logarithmic dependence of the query time bound on $\Delta$ cannot be removed.

- **Lower bounds**: For the other data structure variants studied (NSG, HNSW and DiskANN with "fast preprocessing") we present a family of point sets of size $n$ for all $n$ large enough, such that for each $n$, the *empirical* query time required to achieve "reasonable" accuracy is linear in $n$. For example, for DiskANN, we show that the query procedure can take at least $0.1n$ steps before it encounters any of the 5 nearest neighbors of the query. Remarkably, the point sets are relatively simple: they live in a 2-*dimensional* Euclidean space, and therefore have a *constant doubling dimension* (see Preliminaries). We use implementations provided by the authors, publicly available on GitHub [16, 29, 22]. Our hard instance examples are available on GitHub at [18].

To the best of our knowledge, these are the first worst-case upper bounds and lower bounds for these data structures.

We emphasize that our results do not contradict the empirical evaluations given in the original papers, or in summary studies such as [27]. Indeed, these algorithms have been demonstrated to be very useful and can be highly effective in practice. Nevertheless, we believe that our results provide important information about the behavior of these algorithms. For example, they demonstrate the importance of validating the quality of answers reported by the algorithms when applying them to new data sets. They also shed light on the types of data sets which result in suboptimal performance of the algorithms.

---

[1]E.g., DiskANN has been developed and used at Microsoft and NSG at Alibaba.

[2]Informally, this means that any subset of points that falls into a ball of radius $2r$ can be covered using a constant number of balls of radius $r$. See Preliminaries for the formal definition and discussion.

[3]We note running time bounds that depend on $\log(\Delta)$ versus those that depend on $\log(n)$ are incomparable. Although $\Delta$ could be much larger than $n$, we observed that its value is typically quite low. For example, the MNIST data set [26, 9] contains $n = 60,000$ points, while its aspect ratio $\Delta$ is around 20 when the distances are measured according to the Euclidean norm.

**Related work** Approximate nearest neighbor search has been studied extensively; the references in the introduction provide some of the main milestones of that rich body of research. In the context of this paper, we also mention [25, 33], which studied theoretical properties of modern graph-based nearest neighbor data structure. However, their focus is on the average-case performance, i.e., under the assumption that the data is generated according to some well-defined distribution, like uniform over the sphere. In contrast, this work is focused on the worst-case behavior of those algorithms, and show that their empirical running time can be high even for relatively simple 2-dimensional data sets.

## 2 Preliminaries

We denote the underlying metric space by $(X, D)$. For any point $p \in X$ and radius $r > 0$, we use $B(p, r)$ to denote a ball of radius $r$ centered at $p$, i.e., $B(p, r) = \{q \in X : D(p, q) \leqslant r\}$.

Consider a set of points $P$. We say that $P$ has the *doubling constant* $C$ if for any ball $B(p, 2r)$ centered at some $p \in P$, the set $P \cap B(p, 2r)$ can be covered using at most $C$ balls of radius $r$, and $C$ is the smallest number with this property. Doubling constant is a popular measure of "intrinsic dimensionality" of high dimensional point sets, see e.g., [17, 24, 5]. The value $\log_2 C$ is called the *doubling dimension* of $P$. Doubling dimension is often used as a measure of the "intrinsic dimensionality" of a data set. It generalizes the "standard" (topological) dimension: for any data set $P \subset \mathbb{R}^d$ equipped with a metric $D(p_1, p_2) = \|p_1 - p_2\|_p$, the doubling dimension of $P$ is at most $O(d)$. However, the doubling dimension of $P \subset \mathbb{R}^D$ could be much lower than $D$, e.g., if points in $P$ lie on a low-dimensional manifold. The doubling dimension can be viewed as a finite version of the fractal Hausdorff dimension. Empirical studies (e.g., [11]) showed that the fractal dimension of real data sets is often smaller than their ambient dimension $D$.

The following fact is standard and follows from the definition.

**Lemma 2.1.** *Consider a set of points $P$ with doubling dimension $d$. For any ball $B(p, r)$ centered at some $p \in P$ and a constant $k$, the set $B(p, r) \cap P$ can be covered by a set of $m \leqslant O(k^d)$ balls with diameter smaller than $r/k$, i.e. $B(p, r) \cap P \subseteq \bigcup_{i=1}^{m} B(p_i, r/k)$.*

We use $d$ to denote the doubling dimension of the point set $P$, $\Delta = \frac{D_{max}}{D_{min}}$ to denote the aspect ratio of the input set $P$, where $D_{max}$ ($D_{min}$) is the maximal (minimal) distance between any pair of vertices in the point set. For two points $x_u, x_v$ in $P = \{x_1, ... x_n\}$, we use $D(x_u, x_v)$ to denote the distance between them, or sometimes $D(u, v)$ for simplicity.

## 3 Analysis of DiskANN

In this section we show bounds on the performance of DiskANN with slow preprocessing.

### 3.1 DiskANN recap

In this section we give an overview of the DiskANN procedures. For the full description the reader is referred to the original paper [21], or section A (supplementary material).

The DiskANN data structure is based on a directed graph $G$ over the set $P$, i.e., the set of vertices $V$ of $G$ are associated with the set of points $P$. After the graph is constructed, to answer a given query $q$, the algorithm performs search starting from some vertex $s$. In what follows we describe the search and insertion procedure in more detail.

The search procedure, $GreedySearch(s, q, L)$, has the following parameters: the start vertex $s$, the query point $q$, and the queue size $L$. It performs a best-first-search using a queue of with a bounded length $L$, until the $L$ vertices $v$ with the smallest value of $D(v, q)$ seen so far are all scanned. Upon completion, it returns a list of vertices in an increasing distance from $q$ where the first vertex (or the first $k$ vertices) are answers for the query. Note that as long as the graph is connected, the procedure runs for at least $L$ steps. The total running time of the procedure is bounded by the number of steps times the out-degree bound of the graph $G$.

The construction of the graph $G = (V, E)$ is done by a repeated invocation of a procedure called $RobustPruning$. For any vertex $v$, a set of vertices $U$ (specified later), and parameters $\alpha > 1$ and $R$, $RobustPruning(v, U, \alpha, R)$ proceeds as follows. First, the set $U$ is sorted in the increasing order

of the distance to $v$. The algorithm traverses this sequence in order. After encountering a new vertex $u$, the algorithm deletes all other vertices $w$ from $U$ such that $D(u,w) \cdot \alpha < D(v,w)$. Finally, the node $v$ is connected to all vertices in $U$ that have not been pruned.

The starting point of the DiskANN data structure construction algorithm[4] is the following simple procedure: for each vertex $v$, execute $RobustPruning(v, U, \alpha, R)$ with $U = V$ and $R = n$. That is, robust pruning is applied to *all* vertices in the graph. We refer to this procedure as *slow preprocessing*, as it can be seen that a naive implementation of this method takes time $O(n^3)$. Although the construction time is slow, we show that this construction method provably constructs a graph whose degree depends only logarithmically on the aspect ratio of the graph (assuming constant doubling dimension), and guarantees that the greedy search procedure has polylogarithmic running time. We note that this result is inspired by an observation in [21] about convergence of greedy search in a logarithmic number of steps, though to obtain our result we also need to bound the degree of the search graph and analyze the approximation ratio.

Since the slow-preprocessing-algorithm is too slow in practice, the authors of [21] propose a faster heuristic method to construct the graph $G$, which we call *fast preprocessing* method. At the beginning, the graph $G$ is initialized to be a random $R$-regular graph. Then the construction of the graph $G = (V, E)$ is done incrementally. The construction algorithms make two passes of the point set in random order. For each vertex $v$ met, the algorithm computes a set of vertices $U = GreedySearch(s, x_v, L)$ (for some starting vertex $s$) and then calls the pruning procedure on $U$, not $V$. That is, it executes $RobustPruning(v, U, \alpha, R)$. After pruning is performed, the insertion procedure adds both edges $(v, u)$ and $(u, v)$ for all vertices $u \in U$ output by the prunning procedure. Finally, if the degree of any of $u \in U$ exceeds a threshold $R$, then the set of neighbors of $u$ is pruned via $RobustPruning(u, N_{out}(u), \alpha, R)$ as well. This construction method is implemented and evaluated in the paper.

## 3.2 Analysis: preprocessing

For the sake of simplicity, we let $RobustPruning(p, V, \alpha)$ be the no-degree-limit version of edge selection, i.e., where $R = n$. We first analyze the property of graph constructed by the no-degree-limit pruning, and then show that setting $R = O((4\alpha)^d \log \Delta)$ yields equivalent results.

**Definition 3.1** ($\alpha$-shortcut reachability)**.** Let $\alpha \geqslant 1$. We say a graph $G = (V, E)$ is $\alpha$-shortcut reachable from a vertex $p$ if for any other vertex $q$, either $(p, q) \in E$, or there exists $p'$ s.t. $(p, p') \in E$ and $D(p', q) \cdot \alpha \leqslant D(p, q)$. We say a graph $G$ is $\alpha$-shortcut reachable if $G$ is $\alpha$-shortcut reachable from any vertex $v \in V$.

First, we show that the slow preprocessing algorithm constructs a graph that is $\alpha$-reachable.

**Lemma 3.2** ($\alpha$-shortcut reachable)**.** *For each vertex $p$, if we connect $p$ to the output of $RobustPruning(p, V, \alpha)$, then the graph formed is $\alpha$-shortcut reachable.*

*Proof.* By Definition 3.1, we only need to prove that the constructed graph is $\alpha$-shortcut reachable from each vertex. Suppose that, for some vertex $p$, vertex $q$ is not connected to $p$. Then, in $RobustPruning(p, V, \alpha)$, there must have existed a vertex $p' \in V$ connected to $p$ s.t. $D(p', q) \leqslant D(p, q)/\alpha$. □

Next, we show that the graph produced by no degree limit $RobustPruning$ is actually sparse.

**Lemma 3.3** (sparsity)**.** *For any vertex $p$, let $U = RobustPruning(p, V, \alpha)$, then $|U| \leqslant O((4\alpha)^d \log \Delta)$ where $d$ is the doubling dimension of the point set $P$.*

*Proof.* We use $Ring(p, r_1, r_2)$ to denote the set of vertices that lie in $B(p, r_2)$ but not in $B(p, r_1)$. We use $D_{max}$ ($D_{min}$) to denote the maximal (minimal) distances between a pair of vertices in the point set $P$ and by definition $\Delta = \frac{D_{max}}{D_{min}}$. For each $i \in [\log_2 \Delta]$, we consider $Ring(p, D_{max}/2^i, D_{max}/2^{i-1})$ separately. We cover $Ring(p, D_{max}/2^i, D_{max}/2^{i-1})$ using balls with radius $D_{max}/\alpha 2^{i+1}$ by Lemma 2.1. The number of balls required is bounded by $O((4\alpha)^d)$. Because every two points in the same ball have distance at most $D_{max}/\alpha 2^i$ from each other, and this value is $\alpha$

---

[4]See the discussion in Section 2.2 of [21].

times smaller than their distance lower bound to $p$, at most one of them will remain after performing $RobustPruning(p, V, \alpha)$. Therefore, the number of vertices remain after performing $RobustPruning(p, V, \alpha)$ is upper bounded by $O((4\alpha)^d \log \Delta)$. □

### 3.3 Analysis: query procedure

We now show that if the graph is constructed using the slow indexing algorithm analyzed in the previous section, then $GreedySearch(s, q, 1)$ starting from any vertex $s$ returns an $(\frac{\alpha+1}{\alpha-1})$-approximate nearest neighbor from query $q$ in a logarithmic number of steps.

**Theorem 3.4.** *Let $G = (V, E)$ be an $\alpha$-shortcut reachable graph constructed using the slow preprocessing method. Consider $GreedySearch(s, q, L)$ starting with any vertex $s \in V$ and $L = 1$ (i.e., the algorithm performs no back-tracking) answering query q. The algorithm finds an $\left(\frac{\alpha+1}{\alpha-1} + \epsilon\right)$-approximate nearest neighbor in $O\left(\log_\alpha \frac{\Delta}{(\alpha-1)\epsilon}\right)$ steps.*

*Proof.* Let $a$ be the nearest neighbor of $q$, $v_i$ be the $i$-th scanned vertex, $d_i = D(v_i, q)$, with approximate ratio $c_i = \frac{d_i}{D(a,q)}$. We use $\Delta = \frac{D_{max}}{D_{min}}$ to denote the aspect ratio of the vertex set.

We know that the distance between $v_i$ and $a$ is no more than $d_i + D(a, q)$ by triangle inequality, and because $G$ is $\alpha$-shortcut reachable, each $v_i$ is either connected to $a$, or $v_i$ is connected to a vertex $v'$ whose distance to $a$ is shorter than $\frac{d_i + D(a,q)}{\alpha}$. In best-first-search, the next scanned vertex $v_{i+1}$ should have distance from $q$ no farther than $v'$, which is at most $\frac{d_i + D(a,q)}{\alpha} + D(a, q)$. By induction:

$$d_i \leqslant \frac{D(s,q)}{\alpha^i} + \frac{\alpha+1}{\alpha-1}D(a, q) \tag{1}$$

Consider the following three cases, depending on the value of $D(a, q)$:

Case (1): $D(s, q) > 2D_{max}$. In this case we have $D(a, q) > D(s, q) - D(a, s) > D(s, q) - D_{max} > D(s, q)/2$. Plugging this into inequality 1 gives us $c_i = \frac{d_i}{D(a,q)} \leqslant \frac{D(s,q)}{\alpha^i D(a,q)} + \frac{\alpha+1}{\alpha-1} \leqslant \frac{2}{\alpha^i} + \frac{\alpha+1}{\alpha-1}$. Therefore, for any $\epsilon > 0$, we get a $\left(\frac{\alpha+1}{\alpha-1} + \epsilon\right)$-approximate nearest neighbor in $\log_\alpha \frac{2}{\epsilon}$ steps.

Case (2): $D(s, q) \leqslant 2D_{max}$ and $D(a, q) \geqslant \frac{\alpha-1}{4(\alpha+1)}D_{min}$. By inequality 1, the algorithm reaches an $(\frac{\alpha+1}{\alpha-1} + \epsilon)$-approximate nearest neighbor when $\frac{D(s,q)}{\alpha^i} < \epsilon D(a, q)$. Substituting the value of $D(s, q)$ and $D(a, q)$ with corresponding upper and lower bound, we obtain that the number of steps needed is $\log_\alpha \frac{8(\alpha+1)\Delta}{(\alpha-1)\epsilon} \leqslant O\left(\log_\alpha \frac{\Delta}{(\alpha-1)\epsilon}\right)$

Case (3): $D(s, q) \leqslant 2D_{max}$ and $D(a, q) < \frac{\alpha-1}{4(\alpha+1)}D_{min}$. Suppose at step $i$, $d_i > d(a, q)$ is not the nearest neighbor. Here we obtain a new lower bound for $d_i$. We know $d(v_i, a) > D_{min}$, $d(v_i, q) > d(a, q)$, and $d(a, q) < D_{min}$. By triangle inequality, we have $d_i = d(v_i, q) > d(v_i, a)/2 > D_{min}/2$. Combing this with inequality 1, we obtain that if $v_i$ is not the exact nearest neighbor, it must satisfy $\frac{D_{min}}{2} \leqslant d_i \leqslant \frac{D(s,q)}{\alpha^i} + \frac{\alpha+1}{\alpha-1}D(a, q) \leqslant \frac{2D_{max}}{\alpha^i} + \frac{D_{min}}{4}$. This can only happen when $i \leqslant \log_\alpha 8\Delta$. Therefore, the algorithm reaches the exact nearest neighbor in $O(\log_\alpha \Delta)$ steps. □

We note that similar neighbor selection strategies in the greedy search procedure are also used in HNSW [28] and NSG [15]. However, we cannot prove query performance guarantees similar to the ones above, as those algorithms use $\alpha = 1$.

### 3.4 A tight convergence rate lower bound for DiskANN

In this section we show that the logarithmic dependence of the query time bound on $\Delta$ is unavoidable, i.e., we cannot replace $\log \Delta$ with $\log n$. The proof is deferred to appendix B.

**Theorem 3.5.** *For any choice of $\alpha > 1$, there exists a set of $n = 2k - 1$ points on a one dimensional line with aspect ratio $\Delta = O(\alpha^n)$, and a query q, such that after the slow preprocessing version of DiskANN, greedy search must scan at least $\Omega(\log \Delta)$ or $\Omega(n)$ vertices before reaching an $O(1)$-approximate nearest neighbor of q.*

## 3.5 A tight approximation lower bound for DiskANN

In Theorem 3.4, we know that the slow preprocessing version of DiskANN algorithm can asymptotically get to an $\frac{\alpha+1}{\alpha-1}$-approximate nearest neighbor. Here, we provide a simple instance showing that this approximation ratio is tight.

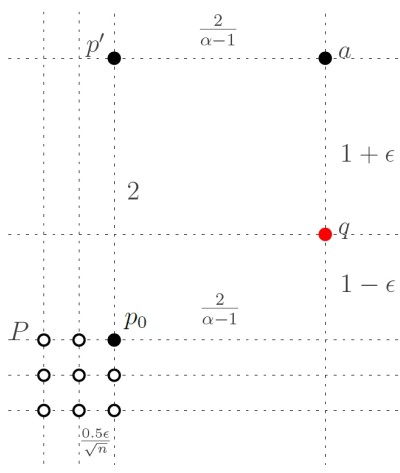

Figure 1: $\frac{\alpha+1}{\alpha-1}$-approximation ratio lower bound instance for slow preprocessing version of DiskANN. The whole instance can be embedded in a two-dimensional grid using $l_1$ distance, and therefore has a constant doubling dimension. Black dots (solid or hollow) are points in the database, the red dot is the query point. **Grids are only used to show the layout structure.** Please refer to Section 3.5 for detailed instance description.

**Theorem 3.6.** *For any $\alpha > 1$, there exists a set of $n + 2$ points in the two dimensional plane under $l_1$ distance where the slow preprocessing version of DiskANN will scan at least $n$ vertices before getting to a vertex with an approximation ratio smaller than $\frac{\alpha+1}{\alpha-1}$-approximate nearest neighbor.*

We draw the instance in Figure 1. The whole instance can be embedded in a two dimensional grid using $l_1$ distance. Note that in the figure, grids are drawn to label distances and highlight the layout structure. The vertex set $V$ consists of a point set $P$ and two single points $p'$ and $a$. The point set $P$ further consists of a $\sqrt{n} \times \sqrt{n}$ (we assume that $n$ is a perfect square) square grid with grid length $\frac{0.5\epsilon}{\sqrt{n}}$ where each grid point is associated with a point. We denote the upper-rightmost point in $P$ by $p_0$. Some important distances are $D(p_0, p') = 2$, $D(p', a) = \frac{2}{\alpha-1}$, $D(p_0, a) = \frac{2\alpha}{\alpha-1}$, $D(a, q) = 1 + \epsilon$, $D(p_0, q) = \frac{2}{\alpha-1} + 1 - \epsilon$, where $\epsilon < 0.01$.

**Lemma 3.7.** *Some properties regarding the graph $G = (V, E)$ built on the instance in Figure 1 using the slow preprocessing version of DiskANN:*

(1) *For any $p \in P$, $(p, p') \in E$ and $(p, a) \notin E$*

(2) *The subgraph of $G$ induced by point set $P \subseteq V$ is strongly connected.*

(3) *The starting vertex $s$ is in $P$.*

*Proof.* (1): According to the $l_1$ distance, we can see that for any point $p \in P$, $D(p, p') > D(p_0, p') = 2$ and $D(p, a) > D(p_0, a) = 2 + \frac{2}{\alpha-1} = \frac{2\alpha}{\alpha-1}$, and $D(p', a) = \frac{2}{\alpha-1}$. Therefore, in the procedure $RobustPruning(p, V, \alpha)$, the edge $(p, p')$ remains and edge $(p, a)$ is pruned.

(2): Because the point set $P$ is a uniform grid, we know that for each vertex $p$, the distances between $p$ and its four adjacent vertices in the grid are smaller than to all other vertices and must remain after $RobustPruning(p, V, \alpha)$. Therefore the subgraph induced by the vertex set $P$ in the final graph $G$ is strongly connected.

(3): In the implementation of DiskANN, the starting point is the closest vertex to the centroid of the whole vertex set, which lies in $P$. $\qquad\square$

*Proof of Theorem 3.6.* Now we analyze the behavior of running $GreedySearch(s, q, L)$ on the instance above on the graph constructed by DiskANN's slow version. We will show that the first $n$ vertices entering the queue are all vertices in the set $P$. Additionally, the only neighbor of set $P$, $p'$, is further from $q$ than any vertex in the set $P$. If $L \leq n$, $GreedySearch(s, q, L)$ terminates before getting to $p'$ or $a$. Consequently, the nearest vertex returned by $GreedySearch(s, q, L)$ has distance at least $\frac{2}{\alpha-1} + 1 - \epsilon$ from $q$, which is not a $\frac{\alpha+1}{\alpha-1}$-approximate nearest neighbor when $\epsilon$ approaches 0.

The first vertex in the queue is $s \in P$. According to property (1) in Lemma 3.7, no vertex in $P$ is connected to $a$, any vertex $p \in P$ has distance $D(p, q) < D(p', q)$ thus has higher priority to be scanned than $p'$, and the subset $P$ is strongly connected, so in the first $n$ steps (recall that $|P| = n$), $GreedySearch(s, q, L)$ will always scan vertices in $P$. □

## 4 Experiments

In our experiments, we test the performance of DiskANN, NSG, and HNSW, on two of our constructed hard instances. We run each algorithm on 20 different data sizes $n \in \{10^5, 2 \cdot 10^5, \ldots, 2 \cdot 10^6\}$. Each data set consists of $n$ points in the two-dimensional plane. We plot Recall@5 rates (Figures 3 and 5) and approximation ratios (Figure 6) for answering the query with queue length $L$ equal to $pn$ where the percentage $p$ is enumerated from the set $1\%, 2\%, \ldots, 12\%, 15\%, 18\%, 20\%, 30\%, 40\%, 50\%$. Note that for all graph-based nearest neighbor search algorithms, the value $L$ lower bounds the number of vertices scanned (and therefore the running time) of the algorithm.

Our results show that, unlike on standard benchmarks, the algorithms scan at least $10\%$ of the points in our instances before finding quality nearest neighbors. All experiments were ran on Google Cloud. Since our experiments only use combinatorial measures of algorithm complexity (i.e., the number of vertices searched), the results do not depend on the exact details of the computer architecture.

### 4.1 Hard instance for DiskANN

Though we provide $\left(\frac{\alpha+1}{\alpha-1}\right)$-approximate ratio upper bound for *slow* preprocessing version of DiskANN algorithm, in the experiments, we show that there exists a family of instances where the DiskANN algorithm with *fast* preprocessing cannot reach any top 5 nearest neighbor before scanning $10\%$ of the vertices. We test our constructed instance using the authors' DiskANN code (fast preprocessing) on GitHub [22] . For a comparison, we also test our constructed instance on our implementation of the slow preprocessing DiskANN algorithm.

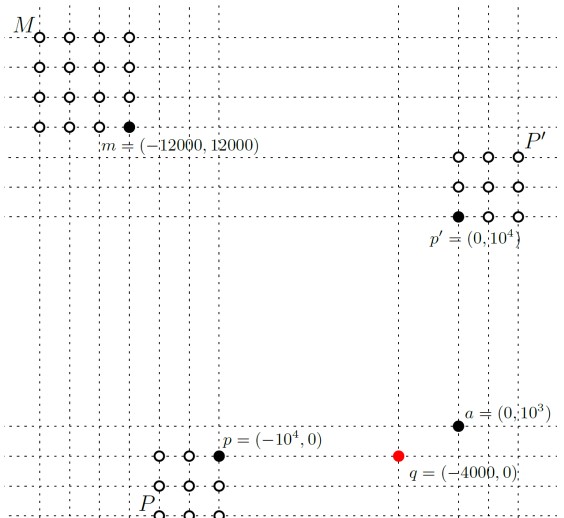

Figure 2: Our constructed hard instance for DiskANN for $n = 10^6$. The instance lives in a two-dimensional Euclidean space, and therefore has a constant doubling dimension. Black dots (solid or hollow) are points in the database, the red dot is the query point. **Grids are only used to show the layout structure.** See section 4.1 for detailed description and supplementary materials for its implementation.

In Figure 2, we draw our constructed instance for the DiskANN algorithm for $n = 10^6$. It is easy to mimic this instance for other data sizes. The parameters used in our experiment are $R = 70$, $L = 125$ (suggested in the original DiskANN paper [21], section 4.1, for in-memory search experiments), $\alpha = 2$, $num\_threads = 1$.

In the left plot of Figure 3, we can see that in most cases, DiskANN (fast preprocessing version) cannot achieve non-zero Recall@5 unless scanning at least $10\%$ of the vertex set. In the right plot of Figure 3, we can see that DiskANN (slow preprocessing version) generates very sparse graph data structure given our hard instance. Furthermore, after preprocessing, greedy search is very efficient: it finds the exact nearest neighbor in **only 2 steps!** Overall, it can be seen that, on the hard instances proposed in this paper, the greedy search procedure for query answering is highly efficient after slow preprocessing, and slow (linear time) after fast preprocessing.

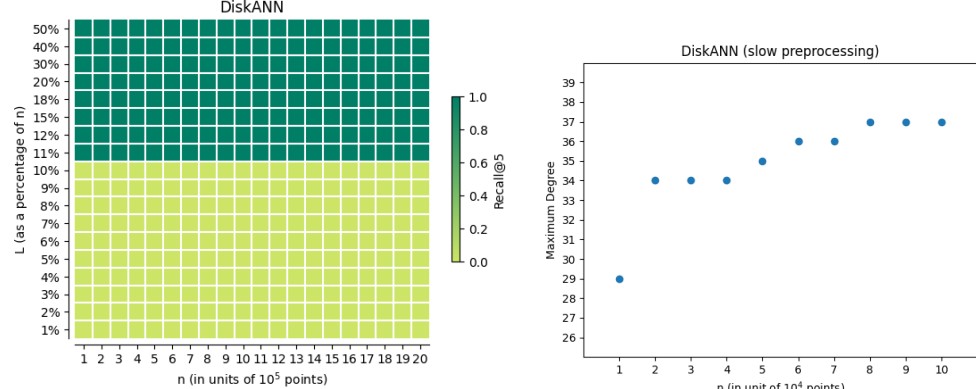

Figure 3: Results for both variants of the DiskANN algorithm on the instance in Figure 2. **Left figure (fast preprocessing DiskANN):** the horizontal axis represents the data size $n$, in multiples of $10^5$ points. The vertical axis represents the size of the search queue length $L$ in terms of the percentage of the data size. Each pixel represents the average value of $Recall@5 \in [0, 1]$ over 10 runs of the algorithm. Note the sharp transation of the recall (from 0 to 1) for $L \approx 10\%$. **Right figure (slow preprocessing DiskANN):** as indicated by Theorem 3.4, slow preprocessing DiskANN constructs a very efficient data structure. In the experiment, it finds the exact nearest neighbor **in at most 2 steps** of greedy search, with $L = 1$. In the figure we plot the maximum degree of the constructed graph (on the vertical axis) vs. the data size $n$ in multiples of $10^4$ points (on the horizontal axis). The plot ends at $n = 10^5$, as the slow preprocessing algorithm takes several hundred hours for larger values of $n$.

**Description of instance in Figure 2**    The instance lives in a 2-dimensional plane under the $l_2$ distance, so in what follows, we give the coordinates for each point, which defines the distances between all points. In Figure 2, $n = 10^6$, the vertex set $V$ consists of three sets of points $M, P$ and $P'$, of sizes $0.8n, 0.1n, 0.1n$ respectively, and a single answer point $a$. Let $l = 0.01 * n = 10^4$. $M$ is a $\sqrt{|M|} \times \sqrt{|M|}$ square grid with grid side length 1 whose bottom-right corner is $m = (-1.2l, 1.2l)$. $P$ is a $\sqrt{|P|} \times \sqrt{|P|}$ square grid with grid side length 1 whose upper-right corner is $p = (-l, 0)$. $P'$ is a $\sqrt{|P'|} \times \sqrt{|P'|}$ square grid with grid side length 1 and whose bottom-left corner is $p = (0, l)$. (We assume that $|M|, |P|, |P'|$ are perfect squares.) The query point is $q = (-0.4l, 0)$, whose nearest neighbor is $a = (0, 0.1l)$. Since our experiments measure $Recall@5$, we add another 4 points very close to $a$, which is not shown in the figure for simplicity.

**Intuition**    First, the start point $s$ should lie in the point set $M$. The three key properties we want to maintain in the construction of the graph are that (1) $s$ is always connected to at least one vertex in the point sets $P$ and $P'$. (2) Except for the points randomly connected to $a$ at initialization, only the points in the set $P'$ can still have edges to $a$ at the end of the construction. (3) In the final graph, at least $L$ vertices in set $P$ are reachable from start point $s$ without passing through any vertex not in $P$. If these three properties hold, $GreedySearch(s, q, L)$ scans only the vertices in $P$ (except for the start point $s$) until it finds a vertex $p_i \in P$ which is randomly connected to $a$ at initialization. In expectation, this requires scanning $\Omega(n/R)$ vertices, and DiskANN won't reach the nearest neighbor $a$ before that. The actual running time of the code is even slower.

## 4.2    Hard instance for NSG and HNSW

A variant of the hard instance from the previous section works for NSG [15] and HNSW [28] algorithms as well. In Figure 4, we draw this instance for $n = 10^6$. It is easy to mimic this instance for other data sizes.

The parameters of NSG are as follows. We set $K = 400, L = 400, iter = 12, S = 15, R = 100$ for EFANNA [12] to construct the KNN graph. We set $L = 60, R = 70, C = 500$ for constructing NSG from KNN graph. These parameters were used by the authors when testing on GIST1M dataset, whose size is similar to ours. The parameters of HNSW are as follows. $ef_{construction} = 200$ (as used in their example code), $M = 64$ (maximal degree limit in their suggested range), $num\_threads = 1$. We test our constructed instance using the authors' code available at [29] (for HNSW) and at [16]

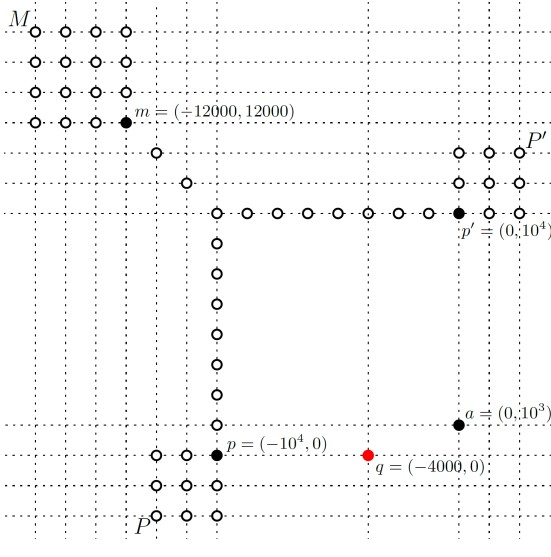

Figure 4: Our constructed hard instance for NSG and HNSW algorithms for data size $n = 10^6$. The instance lives in a two-dimensional Euclidean space, and therefore has a constant doubling dimension. Black dots (solid or hollow) are points in the database, the red dot is the query point. **Grids are only used to show the layout structure.** See Section 4.2 for a detailed description and supplementary materials for its implementation.

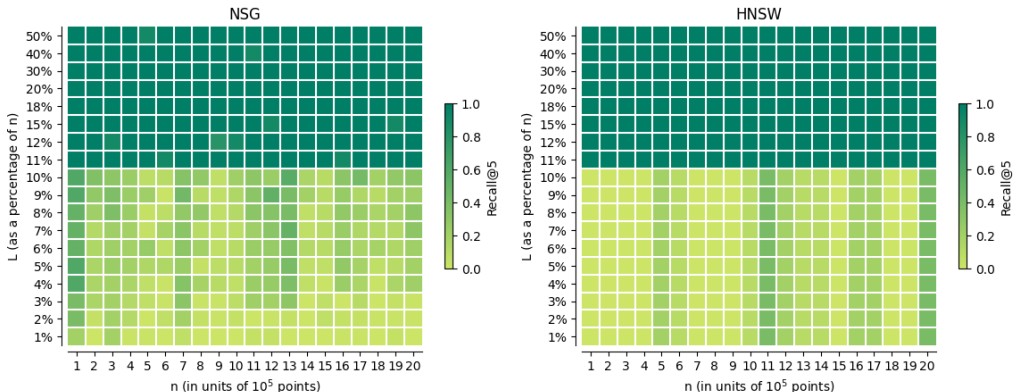

Figure 5: Results for running NSG and HNSW algorithm on instance in Figure 4. The horizontal axis represents the data size $n$, in multiples of $10^5$ points. The vertical axis represents the size of the search queue length $L$ in terms of the percentage of the data size. Each pixel represents the average value of $Recall@5 \in [0, 1]$ over 10 runs of the algorithm. Since the algorithm is randomized, each run generates and uses a different random seed.

(for NSG). We note that both implementations are randomized, but their random seeds are hardwired into the code. To obtain more informative results, we generate and use different random seeds for each algorithm execution, and plot the average recall.

In Figure 5, we can see that, for most values of $n$, both NSG and HNSW algorithms cannot achieve good Recall@5 unless scanning at least $10\%$ of the vertex set.

**Description of instance in Figure 4** This instance is based on the last instance in Figure 2. The main difference is that we use a few chains of points to connect the three subsets $M$, $P$, and $P'$. Specifically, on the dotted line, we add points uniformly spaced out, separated by a distance of 5 in the horizontal and/or vertical directions. For $n = 10^6$ and $l = 0.01n = 10^4$, the instance consists of $400(0.04\%)$ points on the diagonal (from $m = (-1.2l, -1.2l)$ to $(-l, -l)$), $2000(0.2\%)$ on the horizontal line (from $(-l, l)$ to $p' = (0, l)$) and $2000(0.2\%)$ points on the vertical line (from $(-l, l)$ to $p = (-l, 0)$). In total, there are 4400 new added points on the chain compared with the previous instance, occupying $0.44\%$ of the vertex set.

**Intuition for the instance in Figure 4** The new added vertex chains are there to make KNN graph connected for those nearest neighbor algorithms using KNN as part of their constructions. Thanks to these chains, most of the vertices in subset $P$ and $P'$ are reachable from the start point. The construction algorithm will only add edges from the vertices in the set $P'$ to $a$, but not from $P$. Then,

$GreedySearch$ on query $q$ will first traverse the whole subset $P$ before going to $P'$. Therefore, $GreedySearch$ cannot get to the nearest neighbor if the queue length limit $L$ is smaller than $|P|$.

### 4.3 Cross-comparisons

We also evaluated DiskANN on hard examples for NSG and HNSW, and vice versa. For $n = 10^6$ the results are similar to the ones reported in earlier sections: none of the algorithms can achieve non-zero recall unless $L$ exceeds $10\%$ of the data set size. We did not experiment with other values of $n$, as two different families of instances are anyway helpful for other algorithms, as outlined in the next section and described in detail in supplementary material.

### 4.4 Evaluating approximation ratio of the algorithms

In addition to $Recall@5$, we also measure the *average approximation ratio* of DiskANN (fast preprocessing), NSG and HNSW on slightly modified hard instances from Figure 2 and 4. The modifications only involve scaling of the entire instance and bringing points $a$ and $q$ closer, to amplify the approximate ratio for other points. As per Figure 6, the average ratio is very high unless $L \geqslant 0.1n$. Note that we do not include the plot for DiskANN (slow preprocessing) here because, as shown in Figure 3, that algorithm (with $L = 1$) can find the exact nearest neighbor in two steps.

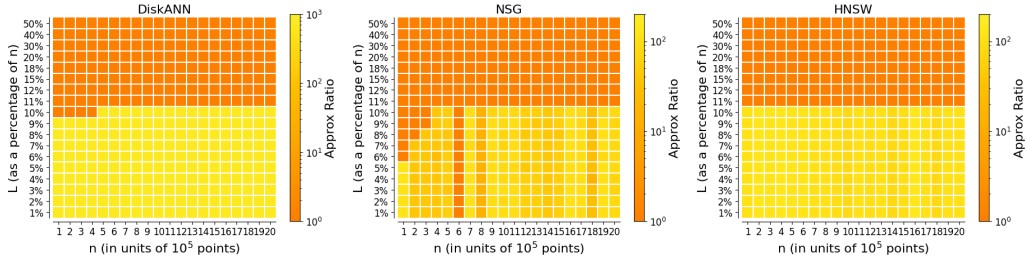

Figure 6: Average approximation ratio results for running DiskANN (fast preprocessing), NSG, and HNSW algorithm on instances in Figure 2 and 4. The horizontal axis depicts the data size $n$, in multiples of $10^5$. The vertical axis depicts the size $L$ of the search queue, as a ratio to $n$. Each pixel represents the average approximation ratio over 10 runs of the algorithm. Since the algorithms are randomized, each run generates and uses a different random seed.

### 4.5 Experiments on other popular approximate nearest neighbor search algorithms

We also tested other popular approximate nearest neighbor search algorithms covered in the survey [37]: NGT [20], SSG [14], KGraph [37], DPG [27], NSW [30], SPTAG-KDT [7] and EFANNA [12]. We ran them on our hard instances, with $n$ ranging over $10^5, 2 \cdot 10^5 \ldots 2 \cdot 10^6$ and the queue length $L = 0.1n$. The table depicts average Recall@5 rates over all values of $n$ and 5 or 10 repetitions. One can see that all algorithms achieve sub-optimal recall. See the appendix for more details.

| DiskANN | NSG | HNSW | NGT | SSG | KGraph | DPG | NSW | SPTAG-KDT | EFANNA |
|---|---|---|---|---|---|---|---|---|---|
| 0.0 | 0.27 | 0.1 | 0.05 | 0.16 | 0.42 | 0.37 | 0.02 | 0.02 | 0.12 |

Table 1: Average Recall@5 for the 10 algorithms surveyed in [37], on our constructed instances.

## 5 Conclusions

In this paper we study the worst-case performance of popular graph-based nearest neighbor search algorithms. We demonstrate empirically that almost all of them suffer from linear query times on carefully constructed instances, despite being fast on benchmark data sets [37]. The exception is DiskANN with slow-preprocessing, for which we bound the approximation ratio and the running time. However, its super-linear preprocessing time makes it difficult to use for large data sets.

An important question raised by our work is whether there is a fast preprocessing algorithm, and a query answering procedure, that are *empirically fast* (e.g., as fast as for DiskANN) while having *worst case* query time and approximation guarantees. Another interesting direction is to investigate whether it is possible to replicate our findings for *real* data sets, using adversarially selected queries.

**Acknowledgement:** This work was supported by the Jacobs Presidential Fellowship, the NSF TRIPODS program (award DMS-2022448), Simons Investigator Award and MIT-IBM Watson AI Lab.

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

# A  DiskANN Algorithm

---

**Algorithm 1** RobustPruning$(i, U, \alpha, R)$

---

1: **Input** Vertex $i$, candidate neighbor set $U$, pruning parameter $\alpha$, degree limit $R$(default $R$ is $n$ if not given)
2: **Result** Update $N_{out}(i)$, the set of out-neighbors of $i$
3: $U \leftarrow U \cup N_{out}(i)$
4: $N_{out}(i) \leftarrow \varnothing$
5: **while** $U \neq \varnothing$ and $|N_{out}(i)| < R$ **do**
6: $\quad v \leftarrow \mathrm{argmin}_{v \in U} D(x_v, x_i)$
7: $\quad N_{out}(i) \leftarrow N_{out}(i) \cup v$
8: $\quad U \leftarrow U \backslash v$
9: $\quad U \leftarrow \{v' \in U : D(x_v, x_{v'}) \cdot \alpha > D(x_i, x_{v'})\}$
10: **end while**

---

**Algorithm 2** GreedySearch$(s, q, L)$

---

1: **Input** Graph $G = (V, E)$, seed $s$, query point $q$, queue length limit $L$
2: **Output** visited vertex list $U$
3: $A \leftarrow \{s\}$
4: $U \leftarrow \varnothing$
5: **while** $A \backslash U \neq \varnothing$ **do**
6: $\quad v \leftarrow \mathrm{argmin}_{v \in A \backslash U} D(x_v, q)$
7: $\quad A \leftarrow A \cup N_{out}(v)$
8: $\quad U \leftarrow U \cup v$
9: $\quad$ **if** $|A| > L$ **then**
10: $\quad\quad A \leftarrow$ top $L$ closest vertices to $q$ in $A$
11: $\quad$ **end if**
12: **end while**
13: sort $U$ in increasing distance from $q$
14: **return** $U$

---

---

**Algorithm 3** DiskANN indexing algorithm (with fast preprocessing)

---

1: **Input** Point set $P = \{x_1...x_n\}$, degree limit $R$, queue length $L$
2: **Output** A proximity graph $G = (V, E)$ where $V = \{1..n\}$ are associated with point sets $P$.
3: $G \leftarrow$ randomly sample a $R$-regular graph on vertex set $V = \{1..n\}$
4: $s \leftarrow$ vertex for the point closest to the centroid of $P$
5: **for** $k = 1$ **to** 2 **do**
6: $\quad \sigma \leftarrow$ a random permutation of $[1...n]$
7: $\quad$ **for** $i = 1$ **to** $n$ **do**
8: $\quad\quad U \leftarrow GreedySearch(s, x_{\sigma(i)}, L)$
9: $\quad\quad RobustPruning(\sigma(i), U, \alpha, R)$
10: $\quad\quad$ **for** vertex $j$ in $N_{out}(\sigma(i))$ **do**
11: $\quad\quad\quad N_{out}(j) \leftarrow N_{out}(j) \cup \sigma(i)$
12: $\quad\quad\quad$ **if** $|N_{out}(j)| > R$ **then**
13: $\quad\quad\quad\quad RobustPruning(j, N_{out}(j), \alpha, R)$
14: $\quad\quad\quad$ **end if**
15: $\quad\quad$ **end for**
16: $\quad$ **end for**
17: **end for**

---

---

**Algorithm 4** DiskANN indexing algorithm (with slow preprocessing)

---

1: **Input** Vertex set $P = \{x_1...x_n\}$, parameters: degree limit $R$
2: **Output** A proximity graph $G = (V, E)$ where $V = \{1..n\}$ are associated with point sets $P$.
3: $s \leftarrow$ vertex for the point closest to the centroid of $P$
4: **for** $i = 1$ **to** $n$ **do**
5: $\quad N_{out}(i) \leftarrow RobustPruning(i, V, \alpha, R)$
6: **end for**

---

# B  Dependence on aspect ratio $\Delta$

Consider the following 1-dimensional data set with $n = 2k$ points located at $\{x_i\}_{i=1}^n$, where

$$x_i = \begin{cases} \alpha^i & \text{for } 1 \leqslant i \leqslant k \\ 2\alpha^k + \alpha^k\beta - \alpha^{2k+1-i} & \text{for } k < i \leqslant n \end{cases}$$

and $\beta = \max(\frac{1}{\alpha-1}, \alpha - 1)$. This is a symmetric line starting from $0$ to $(2 + \beta)\alpha^k$. Each point's distance toward the closer endpoint is $\alpha$ times larger than that of the previous point.

**Lemma B.1.** *The graph $G = (V, E)$ built on the above instance using the slow preprocessing version of DiskANN satisfies the following properties:*

    *(1) For any $i \in [k + 1, n]$, $(i, k) \in E$ and $(i, j) \notin E$ for any $j < k$*

    *(2) For any $j < i \leqslant k$, $(i, j) \in E$ if and only if $j = i - 1$*

*Since the $x_i$'s are symmetric, the same properties also hold in the other direction.*

*Proof.* (1): For any $i \in [k + 1, n]$, we can check that no vertex $j$ such that $k < j < i$ can delete $k$ from $i$'s neighborhood, because $\frac{x_j - x_k}{x_i - x_k} > \frac{\alpha^k \beta}{\alpha^k \beta + \alpha^k} \geqslant \frac{1}{\alpha}$. Thus, we have $(i, k) \in E$. Similarly, we have that $k$ will delete any vertex $j < k$ from $i$'s neighborhood because $\frac{x_k - x_j}{x_i - x_j} \leqslant \frac{\alpha^k}{\alpha^k + \alpha^k \beta} \leqslant \frac{1}{\alpha}$. These two inequalities use that $\beta = \max(\frac{1}{\alpha-1}, \alpha - 1)$

(2): For any $i \in [1, k]$, $x_{i-1}$ is the closest point on $x_i$'s left, so $(i, i - 1) \in E$. Then, for any $j < i - 1$, $j$ will be deleted from $i$'s neighbor by $i - 1$ because $\frac{x_{i-1} - x_j}{x_i - x_j} < \frac{\alpha^{i-1}}{\alpha^i} = \frac{1}{\alpha}$. $\qquad\square$

*Proof of Theorem 3.5.* Based on the graph properties in Lemma B.1, let us determine the length of the shortest path from a starting point $s$ (selected arbitrarily by the DiskANN algorithm) to a constant approximate nearest neighbor of a given query $q$. We select our query $q$ to be either $0$ or $2\alpha^k + \alpha^k \beta$, i.e., one of the two endpoints of the data set, whichever is farther from $x_s$. To find an $O(1)$-approximate nearest neighbor of the query $q$ within $l$ steps of GreedySearch, there should be at least one path with less than $l$ hops from $s$ to $q$'s approximate nearest neighbor. WLOG, let's assume $q = 0$ and $s > k$. By Property (1) of Lemma B.1, among $\{1...k\}$, the vertex $k$ is the only neighbor of any vertex on the right of $k$. By Property (2) of Lemma B.1, for any vertex $i \in [1, k]$, its only neighbor on its left is $i - 1$. Therefore, it takes at least $l \geqslant \Omega(n)$ and $l \geqslant \Omega(\log \Delta)$ steps for slow preprocessing DiskANN with GreedySearch to reach any $O(1)$- approximate nearest neighbor in this constructed instance. $\qquad\square$

## C    More experimental results

### C.1    Hard instance for KD-Tree based nearest neighbor search algorithm

Some nearest neighbor search algorithms use KD-tree to find the entry point for greedy search. In this case, we design a hard instance where KD-tree cannot get good entry point close to the nearest neighbor. See Figure 7. We draw our constructed instance for $n = 10^6$. It is easy to mimic this instance for other data sizes.

**Description of instance in Figure 7**    Our instance has 6 dimensions, where the first two dimensions of the instance are similar to Figure 2 but with different choices of parameters. Again, the vertex set $V$ consists of three sets of points $M, P, P'$ (with size $0.1n, 0.1n, 0.8n$ respectively) and a single answer point $a$: $M$ is a $\sqrt{|M|} \times \sqrt{|M|}$ square grid with unit length 1 and with bottom right corner $m = (-10^9, 10^9)$. $P$ is a vertical chain of points consisting of unit intervals from $p_l = (-10^9, -0.05n)$ to $p_h = (-10^9, 0.05n)$. $P'$ is a $\sqrt{|P'|} \times \sqrt{|P'|}$ square grid with unit length 1 whose bottom left corner is $p' = (0, 10^9)$. The answer point is $a = (0, 3 * 10^8)$. The query point is $q = (-3 * 10^8, 0)$. For a vertex $v \in M \cup P' \cup \{a\}$, each of $v$'s other 4 coordinates are sampled from a uniform distribution supported on $[5 * 10^7, 6 * 10^7]$. For a vertex $v \in P$, its other 4 coordinates are sampled from a uniform distribution supported on $[10^7, 2 * 10^7]$. And for point $q$, its other 4 coordinates are all 0. Note that though such choices of parameters are tailored to the implementation of KD-trees used in SPTAG and EFANNA in [37], some general ideas are reusable for constructing hard instances for other implementations.

**Intuition for instance in Figure 7**    The main reason why we construct a new example here is to handle the use of KD-tree to search for a good starting point. Implementation of KD-tree provided in [37] always randomly picks one of the top 5 dimensions with the largest variance as the splitting

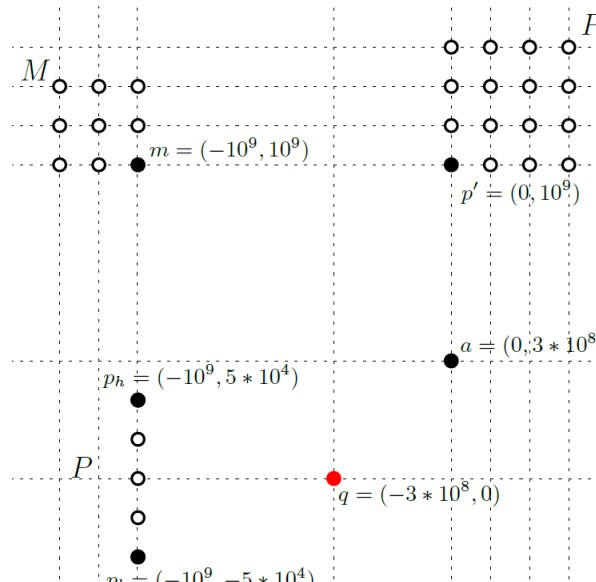

Figure 7: Our constructed hard instance for SPTAG on the scale $n = 10^6$. The instance lives in a two-dimensional Euclidean space, and therefore has a constant doubling dimension. Black dots (solid or hollow) are points in the database, the red dot is the query point. **Grids are only used to show the layout structure.** See section C.1 for detailed description.

dimension, and divides the vertices into two halves at their mean. In Figure 7, because of the separation on the other 4 dimensions, our construction can make sure that KD-tree quickly gets to a subtree with vertices only in $P$. Then KD-tree will only horizontally split the chain from $p_h$ to $p_l$ or split via the other 4 dimensions. In the vertical axis, the coordinates for vertices in $P$ and $a$ are quite close, so KD-tree will assign them a low distance estimation based on only a horizontal split, resulting in KD-tree scanning all vertices on the chain before scanning other vertices outside of the set $P$. As long as we make the KD-tree select a vertex in $P$ as the starting point, we can ensure (as in Figure 2) that $GreedySearch$ will scan all vertices in $P$ before going to $M$ or $P'$.

### C.2 More experiments on other popular nearest neighbor search algorithms

We further test the other 7 popular nearest neighbor search algorithms studied in the survey [37]. We use the same setting as in Section 4. We run each algorithm for 20 different data sizes $n \in \{10^5, 2 \cdot 10^5, \ldots, 2 \cdot 10^6\}$ using hard instances in Figure 4, Figure 2, or Figure 7 (introduced in Appendix C.1). We plot the Recall@5 rate for answering the query with queue length $L$ equal to $pn$ where the percentage $p$ is enumerated from the set $1\%, 2\%, \ldots, 12\%, 15\%, 18\%, 20\%, 30\%, 40\%, 50\%$.

**NGT [20]** We use NGT's implementation from the authors' GitHub repository [19]. We run NGT on the hard instance in Figure 2, using all default parameters as stated in GitHub's readme, except that we use the command "-i g" to generate only the graph index, because of our focus on graph-based nearest neighbor search algorithms. We use command "-p 1" to set the number of threads to 1. We run this experiment 10 times and report the average recall.

**SSG [14]** We use SSG implementation from the authors GitHub repository [13]. We run SSG on the hard instance in Figure 4, using parameters $K = 200, L = 200, iter = 12, S = 10, R = 100$ (for building KNN graph) $L = 100, R = 50, Angle = 60$ (constructing SSG). The parameters chosen here are copied from the author's selected parameters for data set SIFT1M [23], whose data size is close to ours. We run this experiment 10 times using different random seeds and report the average recall.

**KGraph** We use KGraph implementation due to [37], from the GitHub repository [38]. We run KGraph on the hard instance in Figure 4 using parameter $K = 100, L = 130, iter = 12, S = 20, R = 50$. Parameters here are copied from the authors' selected parameters used for their synthetic data set named "$n\_1000000$", whose size is close to ours. We run this experiment 5 times using different random seeds and report the average recall.

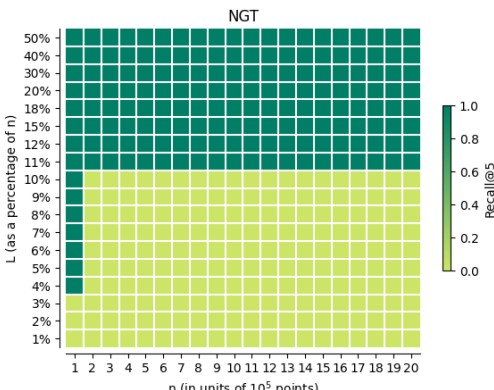

Figure 8: Results for running NGT algorithm on the family of instances in Figure 2. The horizontal axis represents the data size $n$, in multiples of $10^5$ points. The vertical axis represents the size of the search queue length $L$ in terms of the percentage of the data size. each pixel represents a query result, with its $Recall@5 \in [0,1]$ mapping to the spectrum on the right. We run NGT algorithm 10 times and report the average recall rate.

**DPG [27]**  We use DPG implementation due to [37], from the GitHub repository [38]. We run DPG on the hard instance in Figure 4 using parameter $K = 100, L = 100, iter = 12, S = 20, R = 300$. Parameters here are copied from the authors' selected parameters for their synthetic dataset named "$n\_1000000$", whose size is close to ours. We run this experiment 10 times using different random seeds and report the average recall.

**NSW [30]**  We use NSW's implementation due to [37], from the GitHub repository [38]. We run NSW on the hard instance in Figure 4 (with a different vertex permutation) using parameter $max\_m0 = 100, ef\_construction = 400$. Parameters here are copied from the author's selected parameters for their synthetic dataset named "$n\_1000000$", whose data size is close to ours. We run this experiment 5 times using different random seeds and report the average recall.

**SPTATG-KDT [7]**  We use SPTATG-KDT implementation due to [37], from the GitHub repository [38]. We run SPTATG-KDT on the hard instance in Figure 7 using parameters $KDT\_number = 1, TPT\_number = 16, TPT\_leaf\_size = 1500, scale = 2, CEF = 1500$. Parameters here are copied from the authors' selected parameters for their synthetic dataset named "$n\_1000000$", whose size is close to ours. We run this experiment 5 times and report the average recall.

**EFANNA [12]**  We use EFANNA's implementation due to [37], from the GitHub repository [38]. We run EFANNA on the hard instance in Figure 7 using parameter $nTrees = 4, mLevel = 8, K = 80, L = 140, iter = 7, S = 25, R = 150$. Parameters here are copied from the authors' selected parameters for their synthetic dataset named "$n\_1000000$", whose size is close to ours. We run this experiment 10 times and report the average recall.

Experimental results for these 7 algorithms are plotted in Figure 8, Figure 9, and Figure 10. We can see that all algorithms achieve suboptimal Recall@5 rates until the queue length $L$ is greater than 10% of the data size.

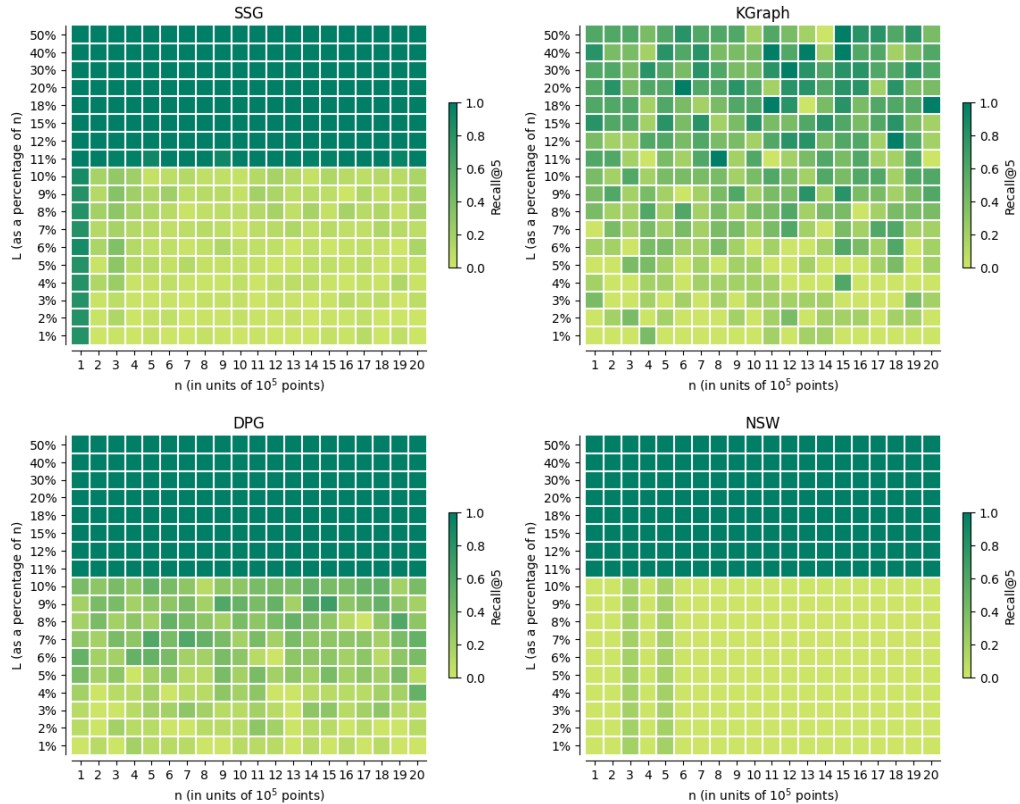

Figure 9: Results for running SSG, KGraph, DPG, NSW algorithms on the family of instances in Figure 4. The horizontal axis represents the data size $n$, in multiples of $10^5$ points. The vertical axis represents the size of the search queue length $L$ in terms of the percentage of the data size. Each pixel represents a query result, with its $Recall@5 \in [0, 1]$ mapping to the spectrum on the right. We run each algorithm 10 (or 5) times and report the average recall rate.

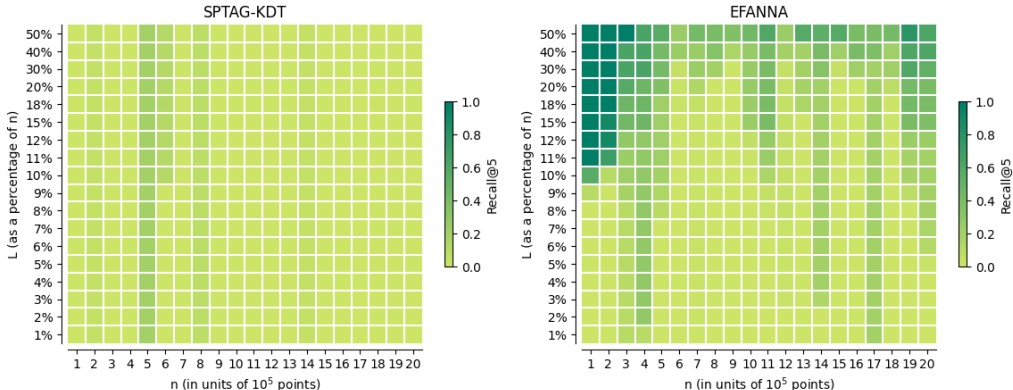

Figure 10: Results for running SPTAG-KDT and EFANNA algorithms on the family of instances in Figure 7. The horizontal axis represents the data size $n$, in multiples of $10^5$ points. The vertical axis represents the size of the search queue length $L$ in terms of the percentage of the data size. Each pixel represents a query result, with its $Recall@5 \in [0, 1]$ mapping to the spectrum on the right. We run each algorithm 10 (or 5) times and report the average recall rate.

