# OpenReview forum: "Worst-case Performance of Popular Approximate Nearest Neighbor Search Implementations: Guarantees and Limitations"
_NeurIPS.cc/2023/Conference — NeurIPS 2023 poster_

### Official Review · Reviewer_GZP3 · 2023-06-22

**Soundness:** 3 good
**Presentation:** 2 fair
**Contribution:** 3 good
**Rating:** 5
**Confidence:** 4

**Summary:**

This paper theoretically analyzes the worst-case performance of DiskANN. It shows that DiskANN (with slow preprocessing) can provably solve the approximate nearest neighbor problem with a constant approximation ratio. It also provides empirical results for other algorithms, such as DiskANN with fast preprocessing, HNSW, and NSG, and shows that there are hard instances that require linear query time.

**Strengths:**

1) The paper theoretically analyzes graph-based nearest neighbor search in quite a general context. While previous works made some strict assumptions about the distribution of elements (uniform), here the analysis is performed in terms of doubling dimension, which is a significant step forward.

2) For one algorithm, DiskANN with slow preprocessing, the constant approximation ratio can be guaranteed. Moreover, it is proven that the theoretically obtained approximation is tight. The proofs are mostly clear and easy to follow.

3) To the best of my knowledge, this is the first paper that shows the effect of neighbor pruning on the performance of graph-based NNS - previous works only considered nearest-neighbor graphs.

**Weaknesses:**

Most of my concerns are about the experimental part:
- Theoretical analysis only shows the constant approximation factor, while the experiments mainly focus on Recall@5 – note that for Recall@5, there are no theoretical guarantees for DiskANN with slow preprocessing.
- It would be informative if all the algorithms were tested on all the “hard” examples and both Recall@5 and approximation ratio were shown. These hard examples may also include the example for DiskANN from Section 3.4. Otherwise, it is hard to conclude that DiskANN is empirically better.
- On the figures illustrating the performance, Recall@5 is shown as a function of L. The figures would be more informative if they showed the fraction of considered nodes (distance computations) instead, as is usually done.
- For a better comparison of the two versions of DiskANN, it would be better to plot Recall@5 as a function of the number of distance computations for both of them.
- Figure 6 analyzes the approximation ratio for three algorithms but not for DiskANN with slow preprocessing.

In summary, currently, it can be hard to conclude the advantage of DiskANN with slow preprocessing over other algorithms since all the algorithms are evaluated in different setups.

Some comments on the presentation:
- Description of DiskANN is important for understanding the theoretical part of the paper. Thus, I suggest moving the key steps of DiskANN to the main text — for instance, the definition of RobustPruning.
- A short description of NSG and HNSW would also help follow their empirical analysis.
- To make a connection with previous works, in Lemma 3.3, it can be useful to comment that for uniformly distributed data, $|U|$ is logarithmic in $n$ and exponential in dimension.
- After Theorem 3.4, it would be helpful to write how the claimed statement follows from (1) and (2).
- Figure 2: $0.5 \epsilon$ should probably be $0.5 \epsilon / \sqrt{n}$ here.

Minor comments:
- I suggest defining the aspect ratio in Section 2 (preliminaries).
- $q$ denotes the query but also some other points in $X$ (e.g., in Sections 2 and 3).
- Proof of Lemma 3.2 is very simple and can be omitted – it can be written that the lemma directly follows from the RobustPruning definition.
- In l136, a reference to Lemma 2.1 can be helpful.
- In Section 4.1, it is better to move the description of the instance to the beginning. Otherwise, it is hard to follow the beginning of the section and Figure 3.
- Parameter R (degree limit) is not discussed in Appendix A (except for the pseudocode).

Some typos:
- l14: “in a some”
- l37, l63: footnotes are properly typeset after punctuation marks
- l70: “they live a 2-dimensional” – missing “in”
- l110: “run” -> “runs”
- l146: “the algorithms performs”
- l153: redundant “for”
- l234: “!.”
- l406: “of with”



**Questions:**

1. In l134, should it be $diam/2^i$ instead of $\Delta/2^i$? Since we want the radius to vary from the minimum distance to diameter.

2. l182 – here a constant bound is given for $\epsilon$, but we also need $\epsilon < 1/(\alpha-1)$ for $a$ to be the nearest neighbor.

3. How specific is Theorem 3.6 to $l_1$ distance? Can similar examples be constructed for $l_2$?

4. Can the analysis be extended to beam search instead of greedy search?

5. Is it possible to guarantee finding the exact nearest neighbor under some conditions? I assume this should be possible if the distance between $q$ and its nearest neighbor is sufficiently small.

6. Could you give more details on how a logarithmic number of steps follows from Theorem 3.4?

7. Can the intuition in the last paragraph of Section 4.1 be potentially transformed to a formal result?

8. Is $d$ assumed to be constant in the paper? Or do all the results hold for $d = d(n)$?

9. I think that it is good to mention that the standard neighbor pruning (used, e.g., in HNSW) uses $\alpha = 1$. Thus, a constant approximation cannot be guaranteed.


**Limitations:**

Limitations are not discussed, the negative societal impact is not relevant for this work.

---

> ### Author Rebuttal · Authors · 2023-08-08
>
> Reviewer GZP3
>
> W1: Theoretical analysis only shows the constant approximation factor, while the experiments mainly focus on Recall@5
>
> A: We believe that our paper treats both measures (Recall@5 and approximation factor) in a fairly balanced way. In particular, for both of the three main algorithms studied (DiskANN, HNSW and NSG) we perform experiments for both Recall@5 and approximation factor. Note that the bulk of sections 4.1, 4.2 and 4.3 is used to describe the hard instances, which are used for both measures. Section 4.3 covers our evaluation of the approximation ratios
>
> W2: It would be informative if all the algorithms were tested on all the “hard” examples and both Recall@5 and approximation ratio were shown. .. Otherwise, it is hard to conclude that DiskANN is empirically better
>
> A: This is an interesting idea and we will perform these experiments for the final version of the paper . We note though that our goal was *not* to demonstrate that the DiskANN is better (or not) on benchmarks, as such a comparison was already made in [19]. Instead our goal was to show that it (the slow-preprocessing version) has strong worst-case guarantees.
>
> As we mention in the paper, we believe that understanding worst case behavior of popular algorithms is very important. For example, they demonstrate the importance of validating the quality of answers reported by the algorithms when applying them to new datasets. They also shed light on the types of data sets which result in suboptimal performance of the algorithms.
>
> W3 & W4: For a better comparison of the two versions of DiskANN, it would be better to plot Recall@5 as a function of the number of distance computations for both of them
>
> A: Our goal for fast-preprocessing DiskANN was to demonstrate linear-time behavior of the search algorithm.  The queue length L lower bounds the number of vertices scanned, which in turn lower bounds the number of distance computations. Thus, the number of distance computations is at least 0.1*n, i.e., 10^4 for n=10^5. .  In contrast, the slow-preprocessing DiskANN search algorithm reported the true nearest neighbor in all cases in just two steps for 10^5 points, so the number of distance evaluations in this case is at most 2*maxdeg<80. This  demonstrates  a large gap between the two variants (on our synthetic examples).
> Unfortunately, performing the slow preprocessing takes lots of time for larger data sets, so we won’t be able to perform more in-depth experiments by the rebuttal deadline. We will, however, perform them for the final version of the paper.
>
> W5: Figure 6 analyzes the approximation ratio for three algorithms but not for DiskANN with slow preprocessing
>
> A: As mentioned on page 7, in the slow preprocessing version of DiskANN, the search algorithm  finds the *exact* nearest neighbor in only 2 steps. Thus, the approximation factor is 1. We will include this note in Figure 6 for clarity.
>
> Q1: In line 134, use $diam/2^i$ instead of $\Delta/2^i$
>
> A:  In the earlier version of the paper, we assumed w.l.o.g. that the distances were scaled so that the diameter was equal to Delta, but we removed this assumption during editing.  Without this assumption,  the radius of the balls should indeed be diameter/2^i instead of  $\Delta/2^i$. Thank you for bringing this to our attention.
>
> Q2: line 182, we also need $\epsilon<1/(\alpha-1)$ for $a$ to be the nearest neighbor
>
> A: Yes, in our proof of theorem 3.6, we consider the case when $\epsilon$ approaches 0, so this constraint is satisfied.
>
> Q3: How specific is theorem 3.6 to l1 distance? Can similar examples be constructed for l2.
>
> A: This is a great question! We began with the l2 norm but found that this straightforward structure doesn’t hold in l2. While it’s possible that challenging instances could exist in the l2 normed space, they might involve greater complexity. To keep things simple, we have chosen to present it using the l1 normed space.
>
> Q4: Can the analysis be extended to beam search instead of greedy search?
>
> A: Our analysis shows that even if the queue length L is equal to 1 (i.e., the search uses a pure greedy algorithm), the algorithm quickly converges to an approximate nearest neighbor. If  L>1 (i.e., when the algorithm performs beam search), the algorithm performance is not worse than for L=1.
>
> Q5: Is it possible to guarantee finding the exact nearest neighbor under some conditions?
>
> A: Yes. For example, the Cover Tree algorithm [5] identifies the exact nearest neighbor in O(log n) time, under a so-called “bounded growth assumption”. Unfortunately,  in general it is not possible to achieve this assuming a bounded doubling dimension, which we consider in this paper.
>
> Q6: Could you give more details on how a logarithmic number of steps follows from Theorem 3.4?
>
> A: The fundamental idea here is that if a point p is not directly connected to q’s nearest neighbor a, then as per RobustPruning outlined in Algorithm 1 on page 12, point p should establish an a alpha-shortcut to a, so each step the distance to the nearest neighbor a is shortened by a factor of alpha.
>
> Q7: Can the intuition in the last paragraph of Section 4.1 be potentially transformed to a formal result?
>
> A: It might be possible to convert this intuition into a formal argument. This would, however, require addressing some subtle probabilistic dependence issues, due to the fact that DiskANN performs multiple passes over the input. Since our goal was to demonstrate a linear running time behavior for specific *implementations* of the studied algorithms, we decided not to pursue this direction in this paper.
>
> Q8: Is d assumed to be constant in the paper? Or do all the results hold for  d=d(n)?
>
> A: Yes, d can be an arbitrary parameter, it does not have to be constant.
>
> Q9: I think that it is good to mention that the standard neighbor pruning (used, e.g., in HNSW) uses $\alpha=1$. Thus, a constant approximation cannot be guaranteed.
>
> A: Thank you for this observation, we'll add it to the paper.

---

> > ### Comment · Reviewer_GZP3 · 2023-08-14
> >
> > I thank the authors for their detailed responses and clarifications! I enjoyed reading the paper and believe that its strengths outweigh its weaknesses. Theoretical analysis in quite a general setup is a strong point. Experiments were a bit confusing for me and the response clarified most of the concerns. If the paper will be accepted, I highly recommend the authors to extend the theoretical part in the main text to make it clear and self-contained. In contrast, the experimental part can be reduced (or partially moved to the supplementary). In particular, the theoretical part of the paper focuses on the approximation ratio, thus the experiments with Recall@5 can be moved to the supplementary material.

---

### Official Review · Reviewer_EYFR · 2023-07-06

**Soundness:** 3 good
**Presentation:** 2 fair
**Contribution:** 2 fair
**Rating:** 6
**Confidence:** 4

**Summary:**

This paper studies a specific class of graph-based similarity search algorithms, and establishes approximation upper bounds, and runtime lower bounds for this algorithm. In the process of establishing lower bounds, the paper also presents various point configurations that would be hard for these graph-based search algorithms, and then evaluates multiple graph-based search algorithms on these hard instances, highlighting that these examples are hard for most search algorithms, even though these graph-based search algorithms are known to perform really well in practice.

**Strengths:**

**Important class of algorithm analysed.**
Graph-based approximate nearest-neighbour search algorithms can be extremely efficient in practice, and the goal of this paper to analyse one such algorithm, DiskANN, is well motivated. This paper establishes upper bounds on the approximation in the nearest-neighbour search result at any given point of the greedy search algorithm. Then, the paper goes on to present explicit examples with interesting geometrical configurations, embedded into a 2 dimensional plane, which are challenging instances of nearest-neighbour search for DiskANN. These examples make explicit one scenario where the graph built on the data forces the greedy search to visit almost all nodes.


**Wide coverage of graph-based search algorithms.**
Both in the empirical evaluation, and the development of "hard instances", this paper covers multiple graph-based search algorithms such as DiskANN, HSNW, NSG, and SPTAG-KDT. Each of these examples exploit the specific graph construction scheme for these methods. The empirical evaluation of these multiple algorithm highlights how these hard instances are hard (to different extents) for all such graph-based algorithms.

**Weaknesses:**

**Interpreting the worst-case upper-bounds.**
One weakness of this paper is that I am unable to get an intuition of what the bound in Theorem 3.4 is telling us. One interpretation is that, as $i \to \infty$ (in the asymptotic range), we get a solution that is $\left(\frac{\alpha+1}{\alpha-1}\right)$-approximate (as stated in line 170-172). But the runtime for $i$ iterations is $O(i (4\alpha)^d \log \Delta)$ (as per Lemma 3.3). But in the nearest-neighbor search, the runtime for the asymptotic $i \to \infty$ scenario is bounded by $O(n)$. So it seems that we are saying that if we wish to achieve constant approximation with DiskANN, we have to do $O(n)$ work, which seems a somewhat vacuous result -- if we are ready to do $O(n)$ work, usually we are able to get the exact solution (not even an approximation). And Theorem 3.5 seems to be saying the same -- we always need to do $O(n)$ work. Does that mean that we are unable to get anything better than $O(n)$ guarantees for DiskANN (even with the slow preprocessing)?

**Interpreting the motivation behind the hard synthetic examples.**
The authors do a great job at creating examples such as the one in Figure 1 & 2, where the slow preprocessing and the fast preprocessing versions of DiskANN respectively are unable to find the constant approximation nearest-neighbor in time less than $O(n)$. Figure 4 is another great synthetic example that is hard for NSG and HNSW. However, it is not directly clear to me why these examples are of interest, or something we should evaluate graph-based algorithms on. These examples are very structured, with very specific careful placements of the query $q$ and its nearest-neighbor $a$. First, it is not clear whether these examples are unique and canonical in some sense, or are there other (similarly constructed) point configurations which would similarly be a hard instance for the graph based algorithms. If they are not unique, why should these examples be studied and not others? Or if we are able to do well on these examples, can we say anything about other "hard" or "easy" instances? Secondly, even if these examples are canonical in some way, it is not clear if these resemble real datasets in any form or if these examples are somehow practical viable. Without such motivation and justification, it is not clear why we should care about these examples.



Minor:
- Without the algorithms being analyzed in the main paper, the analysis is hard to follow. In my opinion, it would useful to have the algorithm being analysed in the main paper.

**Questions:**

- Aspect ratios are somewhat misleading because they sometimes provide counter-intuitive results. For example, searching for near-duplicates (that is, the distance to the nearest-neighbor is close to zero) are arguably the easiest scenario for most nearest-neighbor search methods -- For branch & bound algorithms, the greedy branching usually finds the near-duplicate very quickly, and obtains the tightest possible bound and is able to prune most aggressively. For hashing based methods, near-duplicates almost always collide, hence the search will always find the nearest-neighbor even with a very small number of hash tables. For graph-based algorithms, as long as the graph is well-connected, the greedy search on the graph will get to the near-duplicate node quite quickly and prune everything else after that in the queue. However, bounds based on aspect ratios appear to tell a different story -- near-duplicates cause the aspect ratio to be arbitrarily high, leading to larger upper-bounds on the runtime complexity or the approximation ratio, making it appear that the search problem is harder with near-duplicates. This appears counter-intuitive. How do the results in this paper handle this issue?

- The aspect ratio $\Delta$ used in Lemma 3.3 is based on only the set of points $X$, while the aspect ratio $\Delta$ used in Theorem 3.4 is based on the set of points and the query $q$. So in that case, is it fair to assume that the aspect ratio in Theorem 3.4 **can be** significantly greater than the aspect ratio in Lemma 3.3, especially if (say) $d_{min} \approx 0$.

- In Lemma 3.3, why is it that the index $i$ of the  Rings are restricted to $i \in [\log_2 \Delta]$?

- In Theorem 3.4, what is the range of interest for $i$? As mentioned in lines 170-171, we can "asymptotically get to an $\left(\frac{\alpha+1}{\alpha-1}\right)$-approximate nearest-neighbour". However, it is easy to see that, for different values of $i$, different terms will be the dominating term in $\left( \frac{\Delta}{\alpha^i} + \frac{\alpha+1}{\alpha-1} \right)$, and it would good to understand which values of $i$ we are interested in, and what the guarantees are for those values.

- Can you please provide a motivation for Theorem 3.5? It is quite well known that $O(\log n)$ bounds are hard to get, and usually they come with exponential dependence in dimensionality (in the best case, exponential in the intrinsic dimensionality). Even the celebrated LSH guarantees sublinear bounds in $n$ with polynomial dependence on the dimensionality. So it is generally hard to expect that we can just replace the aspect ratio with the number of points.

- For the experiment in Section 4.4, Table 1, are the results averaged over different "hard instances"? As in, are multiple problem configurations are selected for each of the Figure 1, 2, 4 and 7, and then the average recall@5 is reported across all such problem instances is reported? This point is not clear even in the discussion in Appendix C.2. Furthermore, it would have been good to understand the performances grouped by the hard instances (at least in the appendix). For example, with algorithms such as NSG, DPG, KGraph, where the average is significantly above 0, it would be good to see whether they performed at this mediocre level for all hard instances, or if they performed really well on some hard instances (for example, ones from Figure 2) while struggling on others (for example, figure 7). This information seems important.

**Limitations:**

I did not find any explicit discussion of limitations by the authors. However, I do not anticipate any potential negative societal impact of this work.

---

> ### Author Rebuttal · Authors · 2023-08-08
>
> Reviewer EYFR
>
>
> W1: [interpreting the worst-case upper-bounds] Interpreting the worst-case upper-bounds. One weakness of this paper is that I am unable to get an intuition of what the bound in Theorem 3.4 is telling us. One interpretation is that, as $i\to\infty$ (in the asymptotic range), we get a solution that is  $\frac{\alpha+1}{\alpha-1}$-approximate (as stated in line 170-172). ….
>
> A: We referred to the asymptotic behavior only for simplicity. It can be seen that already for $i=\log_{\alpha} \frac{\Delta}{\epsilon}$ the additive term becomes $\epsilon$, for any $\epsilon>0$. Thus, the convergence to the  $\frac{\alpha+1}{\alpha-1}$-approximation factor is very fast.
>
>
> W2: [interpreting the motivation behind the hard synthetic examples] Interpreting the motivation behind the hard synthetic examples. It is not directly clear to me why these examples are of interest, or something we should evaluate graph-based algorithms on.
>
> A: As we mention in the paper, we believe that our results provide important information about the behavior of these algorithms. For example, they demonstrate the importance of validating the quality of answers reported by the algorithms when applying them to new datasets. They also shed light on the types of data sets which result in suboptimal performance of the algorithms.
>
> In general, understanding the worst-case performance of approximate nearest neighbor methods is a popular subject of study. See e.g., the references in our submission, and a recent paper:
>
> Elkin, Yury, and Vitaliy Kurlin. "A new near-linear time algorithm for k-nearest neighbor search using a compressed cover tree." International Conference on Machine Learning. PMLR, 2023.
>
>
> Q1: Aspect ratios are somewhat misleading because they sometimes provide counter-intuitive results.
>
> A:  We agree. However, as we show in the paper, the logarithmic dependence on $\Delta$ in the running time bound for DiskANN is necessary (Theorem 3.5).
>
> Q2: The aspect ratio $\Delta$ used in Lemma 3.3 is based on only the set of points  X, while the aspect ratio $\Delta$ used in Theorem 3.4 is based on the set of points and the query  q.  So in that case, is it fair to assume that the aspect ratio in Theorem 3.4 can be significantly greater than the aspect ratio in Lemma 3.3,
>
> A:  Indeed, the main result (as stated in the introduction) uses $\Delta$ defined by X and q, while the result stated in Lemma 3.3 uses $\Delta$ defined by X, which could be smaller.  Apologies for the confusion, we will use two different symbols to denote these two quantities.
>
> Q3: In Lemma 3.3, why is it that the index i of the Rings are restricted to i belongs to $i\in [\log_2 \Delta]$?
>
> A: In the earlier version of the paper, we assumed w.l.o.g. that the distances were scaled so that the diameter was equal to $\Delta$, but we removed this assumption during editing.  Without this assumption,  the radius of the balls should be diameter/2^i instead of  $\Delta/2^i$. Thank you for bringing this to our attention.
>
> Q4: in theorem 3.4, what is the range of interest for i?
>
> A: See answer for W1 above.
>
> Q5: Can you please provide a motivation for Theorem 3.5?
>
> A: The theorem shows that  the logarithmic dependence on $\Delta$ cannot be avoided. In contrast, some algorithms achieve O(log n) query time, though under stronger assumptions about the input (bounded growth assumption).  See e.g., [5] and the Elkin - Kurlin paper listed earlier.
>
> Q6: For the experiment in Section 4.4, Table 1, are the results averaged over different "hard instances"?
>
> A: For Table 1 in section 4.4, the results for each algorithm are obtained by running it on its specific instance and averaged over all values of n with 5 or 10 repetitions. For every algorithm, it is executed solely on one specific hard instance (Figure 2, 4, or 7). Please refer to their figure captions to see which instance it is run on. Our objective is to analyze the worst case performance for each algorithm, leading us to construct distinct challenging instances tailored to each algorithm.

---

> > ### Comment · Reviewer_EYFR · 2023-08-21
> >
> > Thank you for your response. I will increase my score. My only follow-up is that W2 was not asking for the motivation for analyzing the worst-case (that is always important as the authors notes). My point about the W2 is that the worst case examples are very structured, and I was asking whether such structured worst case scenarios can be practically motivated.

---

> > > ### Author Response · Authors · 2023-08-21
> > > **Thank you for the response and increasing the score!**
> > >
> > > Regarding the issue of “whether such structured worst case scenarios can be practically motivated”: since graph-based nearest neighbor search algorithms generally perform well in practical scenarios, we believe that real-world situations may not be as unfavorable as the worst-case scenario in our paper indicates. However, our construction is motivated by a simple idea, which is to trap greedy search in a local minimum. Therefore, similar scenarios should exist in practice, but the penalty might not be as severe. This variability in the penalty could contribute to the variance in running times when graph-based nearest neighbor search algorithms are used to answer different queries.

---

### Official Review · Reviewer_ZVW4 · 2023-07-07

**Soundness:** 3 good
**Presentation:** 2 fair
**Contribution:** 3 good
**Rating:** 5
**Confidence:** 2

**Summary:**

Nearest neighbour (NN) queries select for a given data point the closest point in a set and can be answered exactly in linear time by scanning the whole set. c-approximate Nearest Neighbour (ANN) queries ask for an arbitrary data point that has at most a c times larger distance than the NN. A popular approach in the literature is DiskANN that builds as an index structure a proximity graph that links close-by points together. In this manuscript two variants of DiskANN are analysed.

**Strengths:**

S1. The work seems to fill a gap left by prior works: works proposing the studied methods seem to lack a more extensive theoretical analysis as featured in this work.

S2. Constructed problem instances used in experiments provided in supplementary material (as well as links to code of studied ANN approaches) and details of how various methods have been implemented are extensively documented in the supplementary material.

S3. Combination of empirical and theoretical results to furthen the understanding of the worst-case / average-case performance of ANN methods.

**Weaknesses:**

W1. Some parts of the paper are a bit confusing.

It is not clear which point is meant to be the (exact) nearest neighbour (NN) in Figure 1. If "a" is the NN, then it would seem that "p0" would be an approximate nearest neighbour given the values.

**Questions:**

Q1. What is meant to be the (exact) nearest neighbour (NN) of "q" in Figure 1? If the NN of "q" is meant to be "a", why would "p0" not be a c-approximate nearest neighbour with c = (alpha+1)/(alpha-1)?

Q2. If a point that coincides with the query point Q is in the data set, do all c-approximate nearest neighbours (for c > 1) then coincide with the (exact) nearest neighbours? What are the implications of this edge case on the claims in the paper?

**Limitations:**

Relevant limitations seem to be discussed.

---

> ### Author Rebuttal · Authors · 2023-08-08
>
> Reviewer ZVW4
>
> W1: It is not clear which point is meant to be the (exact) nearest neighbor (NN) in Figure 1. If "a" is the NN, then it would seem that "p0" would be an approximate nearest neighbor given the value
>
> A: In our figure 1, q represents the query point and a represents the true nearest neighbor. In the proof of Theorem 3.6, when $\epsilon$ approaches 0, the approximation ratio of p0 is no better than $\alpha+1/\alpha-1$. We will modify the statement of Theorem 3.6’s statement to “before getting to a vertex with an approximation ratio *smaller* than $\alpha+1/\alpha-1$”. Thank you for your observation!
>
> Q2: If a point that coincides with the query point Q is in the data set, do all c-approximate nearest neighbors (for c > 1) then coincide with the (exact) nearest neighbors? What are the implications of this edge case on the claims in the paper?
>
> A: Yes. In this case the distance to the nearest neighbor is zero, so an approximate nearest neighbor algorithm must report the exact nearest neighbor. We believe this fact does not affect the claims in the paper, but if there are any specific concerns, please do let us know.

---

> > ### Comment · Reviewer_ZVW4 · 2023-08-18
> >
> > Thank you, that clarifies Q2 and I think that answers Q1 as well, but have to still revisit the paper and double-check.

---

### Author Rebuttal · Authors · 2023-08-08

We thank all reviewers for their useful comments and feedback. We will fix the typos and presentation issues in the final version of the paper. In what follows we address the issues identified by the reviewers as weaknesses and/or listed as questions.

---

### Decision · Program_Chairs · 2023-09-21

**Decision:**

Accept (poster)

**Comment:**

The paper focuses on theoretical worst case guarantees of popular graph based KNN algorithms. It provides good insight into why these algorithms work and shows hard instances where the algorithms does not work. They verify their findings with empirical experiments.  The reviews were generally positive but lack the excitement. Rebuttal pushed the paper in generally positive direction. The area chair went over the paper. In the end, the papers analysis even if worst case and may not reflect the general success of the paper, will still be useful and eye opening for the community as it points out important things to watch for in some of the most widely used algorithms.